# Augmented Reality Demonstrations for Scalable Robot Imitation Learning

### Yue Yang
yygx@cs.unc.edu
The University of North Carolina at Chapel Hill
Chapel Hill, NC, USA

### Bryce Ikeda
bikeda@cs.unc.edu
The University of North Carolina at Chapel Hill
Chapel Hill, NC, USA

### Gedas Bertasius
gedas@cs.unc.edu
The University of North Carolina at Chapel Hill
Chapel Hill, NC, USA

### Daniel Szafir
dszafir@cs.unc.edu
The University of North Carolina at Chapel Hill
Chapel Hill, NC, USA

## ABSTRACT

Robot Imitation Learning (IL) is a widely used method for training robots to perform manipulation tasks that involve mimicking human demonstrations to acquire skills. However, its practicality has been limited due to its requirement that users be trained in operating real robot arms to provide demonstrations. This paper presents an innovative solution: an Augmented Reality (AR)-assisted framework for demonstration collection, empowering non-roboticist users to produce demonstrations for robot IL using devices like the HoloLens 2. Our framework facilitates scalable and diverse demonstration collection for real-world tasks. We validate our approach with experiments on three classical robotics tasks: reach, push, and pick-and-place. The real robot performs each task successfully while replaying demonstrations collected via AR.

## CCS CONCEPTS

• **Computing methodologies → Imitation Learning**; • **Software and its engineering → Augmented Reality**.

## KEYWORDS

Augmented Reality, Imitation Learning

**ACM Reference Format:**
Yue Yang, Bryce Ikeda, Gedas Bertasius, and Daniel Szafir. 2024. Augmented Reality Demonstrations for Scalable Robot Imitation Learning. In *Proceedings of the 7th International Workshop on Virtual, Augmented, and Mixed-Reality for Human-Robot Interactions at HRI 2024 (VAM-HRI'24)*. ACM, New York, NY, USA, 4 pages. https://doi.org/XXXXXXX.XXXXXXX

## 1 INTRODUCTION

Recent advancements in robot learning have showcased its potential across various manipulation applications [2–4]. While Reinforcement Learning (RL) has emerged as a common approach for developing robot controllers, defining reward functions to elicit desired behaviors can be challenging, potentially leading to overfitting to specific RL algorithms [1] and sample inefficiency [20]. In contrast, Imitation Learning (IL) aims to empower end-users to teach robots skills and behaviors through demonstrations, showing promising results in controlled laboratory environments [3, 4, 11, 13]. However, current methods for collecting demonstrations require users to be acquainted with the operation of specific controllers or engage in contact-based kinesthetic teaching on real robot arms [6, 16, 18], thereby impeding the widespread application of IL.

To streamline demonstration collection for non-roboticists, it is crucial to address two issues: 1) non-expert users typically lack understanding of robot arm controllers, and 2) non-roboticists may face limited access to real robot arms due to their high cost and the specialized nature of robot manipulators. While virtual reality (VR) has been used for addressing these challenges [8, 21], it requires creating VR environments in advance, limiting task diversity. To circumvent this, Duan et al. [5] propose using augmented reality (AR) for visual-input demonstration collection. However, many powerful imitation learning methods still rely on demonstrations in low-dimensional state spaces, such as robot arm joints and end-effector poses [7, 14, 15], rather than utilizing demonstrations in high-dimensional state spaces, such as visual information like images. Additionally, imitation learning often demands many more high-dimensional visual demonstrations [9, 17] compared to low-dimensional demonstrations, increasing user burden. Therefore, collecting low-dimensional demonstrations while addressing the two posed challenges still requires further investigation.

We propose a novel framework that enables non-roboticists to use AR to easily collect demonstrations in low-dimensional state space. Our framework enables users to perform tasks using their hands as they would in daily life, addressing the first challenge. Leveraging AR, our method circumvents the need for real robot arms, solving the second challenge. Our contributions are two-fold:

(1) We present a framework allowing non-roboticists wearing AR glasses to easily collect demonstrations in low-dimensional state space using their own hands.
(2) We deploy our framework on the HoloLens 2 AR platform and assess the gathered demonstrations using a real Fetch robot. The robot successfully completed three sample tasks when replaying collected demonstrations, underscoring the high quality of the demonstrations collected via AR.

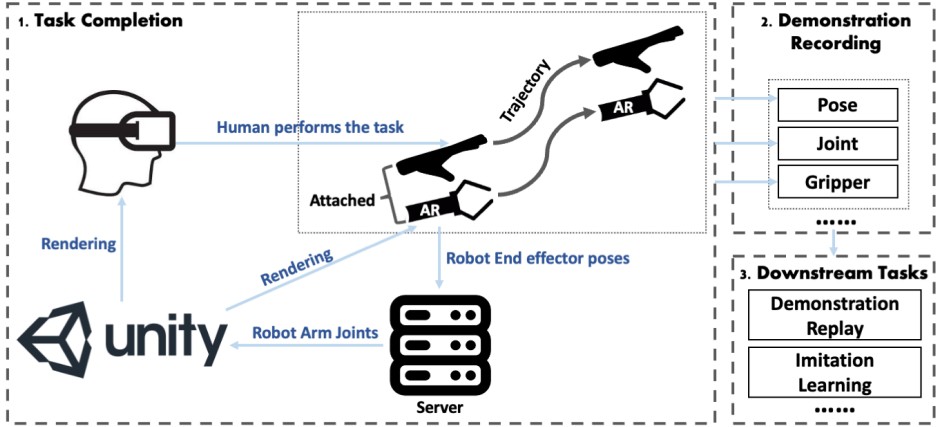

**Figure 1: This figure illustrates the proposed framework's pipeline. The user with AR glasses performs the task while being mindful of the attached AR robot end effector. We capture the low-dimensional states as demonstrations during this process. The collected demonstrations can be readily applied to various downstream tasks, such as demonstration replay and training imitation learning algorithms.**

## 2 METHODOLOGY

We describe the proposed framework, as depicted in Figure 4, through two distinct steps. In Section 2.1, we delineate the process by which users collect demonstrations aided by AR. Subsequently, in Section 2.2, we detail how we enhance the smoothness of collected demonstrations with the assistance of our designed key-poses detection.

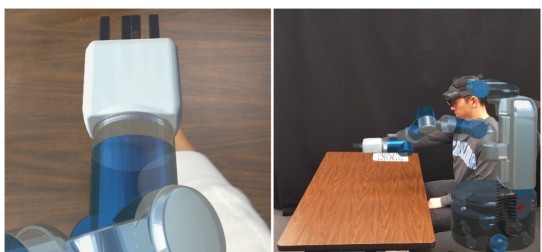

**Figure 2: Left: Egocentric view showing the overlap between the human hand and the robot end effector. Right: The human performs the task manually, with the robot end effector mirroring the hand movements.**

## 2.1 AR-assisted Demonstration Collection

To collect demonstrations of arm trajectories from users, we utilized the Microsoft HoloLens 2, an augmented reality head-mounted display (ARHMD). The HoloLens 2 provides developers with data from eye tracking, a microphone, an inertial measurement unit, and hand tracking. The HoloLens provides both the position and orientation of the wearer's hands, enabling us to map the user's hand movement to the robot directly. During the demonstration process (as seen in Figure 2), users wear the ARHMD, which overlays a digital twin of the robot on the user and provides an egocentric view of the robot's perspective. This setup facilitates real-time visual feedback of the robot's movements to the user. Using current

inverse kinematics algorithms [12], we compute the robot's joint angles based on the demonstrator's hand position. Furthermore, to detect whether a user is picking or placing an object, we calculate the distance between their pointer finger and thumb as they grab a bounding box. If the distance exceeds a specified threshold, it indicates an object being picked up; otherwise, it signifies an object being placed down. To provide accurately positioned visualizations, we align the coordinate frames of the Unity scene and the HoloLens using a fiducial marker placed on the table [10]. As the joint angles, end-effector positions, and pick or place actions are calculated during the demonstrations, this information is recorded on a separate machine, transmitted from the HoloLens via Unity Robotics Hub's ROS-TCP-Endpoint and ROS-TCP-Connector [19]. With this setup, we record one demonstration $D$ with a form as shown in Equation 1. Each demonstration includes $N$ data points, with the form of end-effector pose, $p_i$, corresponding robot arm joints, $j_i$, and binary gripper state, $g_i$ for each timestep $i = 1, ..., N$.

$$D = \{(p_i, j_i, g_i)\}_{i=0}^{N-1} \tag{1}$$

## 2.2 Demonstration Process

The user can collect demonstrations using the ARHMD, but the recordings tend to be jerky for two reasons. First, high recording frequencies are essential to capture dense trajectories. However, these frequencies can also inadvertently record minor hand tremors. Second, inherent inaccuracies in hand-tracking contribute to noise. Applying a filter to the raw demonstrations can smooth out these issues and improve demonstration quality.

We employ a straightforward trajectory downsampling method, retaining every $K$th data point from the collected demonstration. However, to ensure critical data points necessary for task completion are not inadvertently filtered out, we introduce an automatic *key data point* detector. Our approach identifies key data points based on significant angle changes in the user's hand trajectory coupled with slow movement (i.e., approaching zero velocity) or

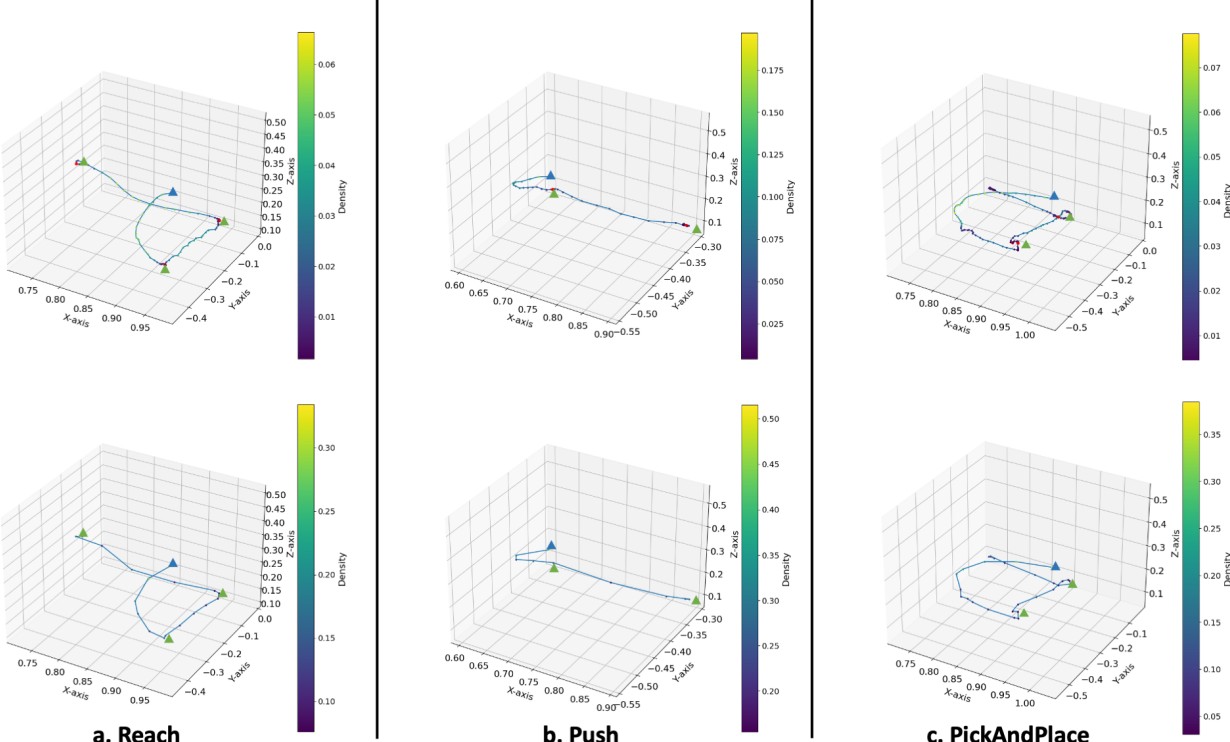

**a. Reach**   **b. Push**   **c. PickAndPlace**

**Figure 3: Each column represents a distinct task. In the upper figure of each task, the visualization displays the demonstration without processing, with red points indicating detected key data points. The lower figure illustrates the demonstration after processing. In all tasks, the blue triangle denotes the initial position of the robot arm's end-effector. For the Reach task, green triangles mark the three-goal waypoints. In the Push and Pick-And-Place tasks, green triangles indicate the starting and goal positions for the object.**

during grasping and releasing actions. Although our recording only captures position-based demonstrations without velocity information, the density of neighboring poses effectively serves as a substitute. Algorithm 1 presents the pseudocode for the former case, where $ComputeAngle(\cdot)$ calculates the angle formed by the current point, the window's start point, and the window's endpoint, while $ComputeDensity(\cdot)$ computes the average pairwise distances within the window. As to grasp/release actions, the required information is inherently present in the recorded demonstration.

## 3 EXPERIMENTS

### 3.1 Tasks Description

We assess the effectiveness of our framework through experiments on three fundamental robotic tasks: Reach, Push, and Pick-And-Place. These tasks represent fundamental manipulation actions that can be combined to accomplish more complex tasks. In the Reach task, the robot arm is tasked with reaching three predefined waypoints. The Push task aims to push an object on the desk from a starting waypoint to a designated goal waypoint. In the Pick-And-Place task, the robot arm is required to grasp an object at one location, transport it to another position, and then release it, effectively relocating the object to the desired goal position.

---

**Algorithm 1:** Key Pose Detection Algorithm

**Input** : *points*: positions extracted from collected poses;
         *window_length*: the duration over which we
         compute pose angles and density;
         *sharp_turn_threshold*; *dense_region_threshold*.

1   *sharp_turn_indexes, dense_region_indexes* ← Empty list
2   **for** *idx, point* **in** *enumerate(points)* **do**
3      *angle* ← ComputeAngle(*point, window_length*)
4      **if** *angle > sharp_turn_threshold* **then**
5         *sharp_turn_indexes.append(idx)*

6   **for** *idx, point* **in** *enumerate(points)* **do**
7      *density_score* ←
        ComputeDensity(*point, window_length*)
8      **if** *density_score > dense_region_threshold* **then**
9         *dense_region_indexes.append(idx)*

**Output** : *sharp_turn_indexes* ∩ *dense_region_indexes*

---

### 3.2 Visualization of Collected Demonstration

By plotting the positions of each collected pose in 3D space, we can visualize the trajectory of the gathered demonstration. As shown

in Figure 3, the demonstration trajectory collected via AR exhibits accurate movement for each task with occasional jerky motion. Upon applying the proposed novel key pose detector to the collected demonstrations, critical data points (the red points) are accurately detected. By retaining these key data points during the downsampling process, the resulting demonstration trajectories become smoother.

## 3.3 Real Robot Demonstration Replay

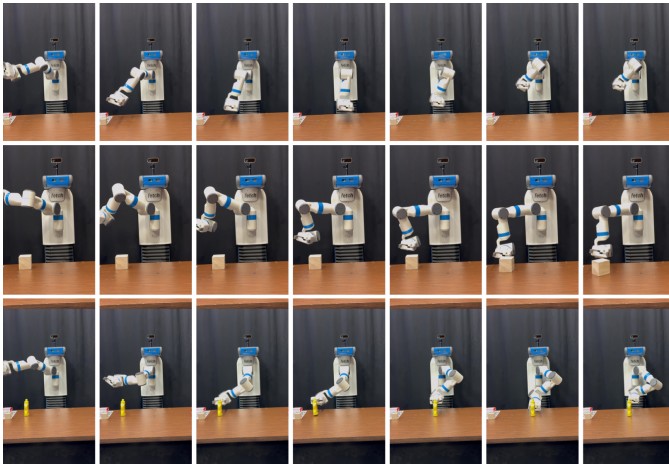

**Figure 4: Each row represents a specific task: Reach, Push, and Pick-And-Place, respectively.**

In each demonstration, we calculate the delta joints between two time steps, yielding a list of joint actions to be applied to the real Fetch robot arm. We leverage ROS and the gym environment to enable the robot arm to execute the generated actions and replay the demonstration. Figure 4 illustrates the successful completion of each task by a Fetch robot, validating our approach toward generating high-quality demonstrations via AR.

## 4 CONCLUSION

We present an AR-assisted demonstration collection framework designed to facilitate scalable demonstration gathering for non-roboticists. By tracking the human hand in an AR environment and employing a novel key data point detector-based demonstration filter, we attain high-quality demonstrations with a user-friendly approach, where users simply perform tasks with their own hands as they normally would. Through visualization and demonstration replay on a real robot we show the quality of the demonstrations collected via our framework and illustrate the validity of our approach.

## 5 ACKNOWLEDGEMENTS

This work was sponsored by the National Science Foundation (NSF) under grant number 2222953.

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
