# OpenReview forum: "Augmented Reality Demonstrations for Scalable Robot Imitation Learning"
_humanrobotinteraction.org/HRI/2024/Workshop/VAM-HRI — VAM-HRI 2024 Oral_

### Official Review · Reviewer_yQhU · 2024-02-26
**Accept**

**Rating:** 9
**Confidence:** 5

**Review:**

This paper presents a lightweight AR-based method for providing demonstrations to be used for robot imitation learning. By tracking the user's hand as a stand-in for a robot end-effector, the system allows users to provide demonstrations in a real-world environment, without requiring a physical robot, specialized input devices, or a built-out simulated environment. The authors describe the system and demonstrate its feasibility using a set of simple robot actions, demonstrated by a user and then visualized in AR, and successfully executed on a real robot.

Strengths:
- This technique of providing demonstrations is very lightweight, requiring only an AR headset as special equipment. As the authors posit, this accessibility makes the collection of large demonstration datasets more feasible, adding to the scalability of robot imitation learning.
- The visualization of the robot end effector mirroring the human hand helps the human user determine whether the demonstration they've given is high quality without requiring execution on a robot, speeding up the process of data gathering.

Weaknesses:
- In order to maintain tracking, the user's hands will need to remain roughly centered in the view of the headset throughout the duration of the trajectory. This might be ergonomically unnatural and would preclude long or complicated tasks that require head movement to maintain awareness.

This paper represents a highly promising approach using AR to simplify the process of providing demonstrations to robots. I look forward to the discussion this work will generate at the workshop, especially in planning upcoming analyses of the system with respect to human factors and trajectory quality. I recommend acceptance to this year's VAM-HRI workshop.

---

### Official Review · Reviewer_GD6E · 2024-02-26
**Review b**

**Rating:** 8
**Confidence:** 5

**Review:**

The paper describes an AR framework for imitation learning where the operator can demonstrate an action in AR to a robot so the robot can learn it and execute it in the future. In genereal the paper is a complete and strong work showing that the (untrainted) user can succesfully train the robot to execute a task.

Even though authors mentione that modeling such a start would be harder in VR, It would be interesting to see how such framwork compares between VR and AR and what users would prefer.
Another addition to this work would be a user study analysis where the authors analyse users' answers and preferences about the framework.

Overall, great work, accept.

---

### Decision · Program_Chairs · 2024-02-26

Accept (Oral)